# S3E: Semantic Symbolic State Estimation With Vision-Language Foundation Models

## Abstract

In automated task planning, state estimation is the process of translating an agent's sensor input into a high-level task state. It is important because real-world environments are unpredictable, and actions often do not lead to expected outcomes. State estimation enables the agent to manage uncertainties, adjust its plans, and make more informed decisions. Traditionally, researchers and practitioners relied on hand-crafted and hard-coded state estimation functions to determine the abstract state defined in a specific task domain. Recent advancements in Vision-Language Models (VLMs) enable autonomous retrieval of semantic information from visual input. We present Semantic Symbolic State Estimation (S3E), the first general-purpose symbolic state estimator based on VLMs that can be applied in various settings without specialized coding or additional exploration. S3E takes advantage of the foundation model's internal world model and semantic understanding to assess the likelihood of certain symbolic components of the environment's state. We analyze S3E as a multi-label classifier, reveal different kinds of uncertainties that arise when using it, and show how they can be mitigated using natural language and targeted environment design. We show that S3E can achieve over 90% state estimation precision in our simulated and real-world robot experiments.

## 1 Introduction

Automated task planning is a crucial component for intelligent agents to solve complex and ever-changing tasks (Ghallab et al., 2016; Geffner & Bonet, 2013). In some cases, it is appropriate to assume that an agent has full domain knowledge (the Closed World Assumption (CWA) (Reiter, 1981)) and that all facts about the world are known. However, an agent's observations are often based on its sensing capabilities, especially in real-world applications, from which extracting these facts is non-trivial. State estimation is the process of obtaining a high-level state of the environment, i.e., translating numeric sensor input into semantic facts. (Chen et al., 2024a; Castaman et al., 2021; Lagriffoul et al., 2018). State estimation is crucial for monitoring the execution of a plan. Should an action lead to an unexpected state, there may be grounds for replanning or reporting of task failure.

**Example 1** *A robotic arm is tasked with rearranging groceries on multiple tables. The goal is to move a box of cereal and a carton of milk to a specific table where the hungry human would like to prepare her breakfast. A task planner chooses the following plan: "pick-up(milk, table1)", "put-down(milk, table3)", "pick-up(cereal, table2)", "put-down(cereal, table3)". While moving to place the cereal on table number 3, the the object is dropped due to a bad grasp, and lands on table number 1. Using a state estimator, we notice that the expected state where the cereal box is on table number 3 has not been achieved. We thus call the task planner once more to obtain the following plan that will lead us to the goal state: "pick-up(cereal, table1)", "put-down(cereal, table3)".*

Current state-of-the-art methods for task planning rely on hand-crafted and hard-coded state estimation functions (Moreno et al., 2024; Garrett et al., 2020). This is time-consuming work that relies on advanced sensing equipment, which results in domain-specific outputs that do not adapt to any changes in the environment or task. We desire a general state estimation function that requires no specialized coding, no additional exploration, and generalizes to a large scope of tasks.

With the rise of powerful instruction-based Vision-Language Models (VLMs), i.e., vision-based foundation models, it is now possible to answer complex semantic questions about a scene based

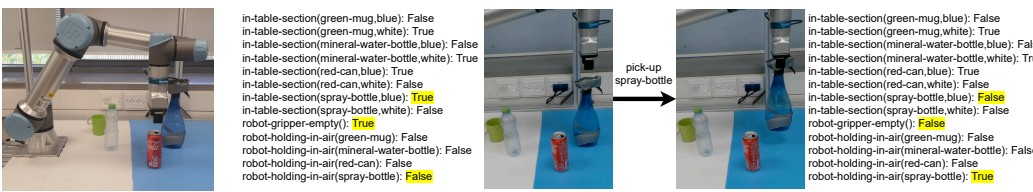

(a) Setup.  (b) Example transition annotated with S3E. State changes highlighted.

Figure 1: Visual results from a robotic pick-and-place task using S3E - after picking up the spray bottle, 'robot-gripper-empty()' and in-table-section(spray-bottle, blue)' are set from True to False. We refer the reader to the supplementary materials for a demo video of this example.

on visual input alone (Liu et al., 2023b; OpenAI, 2023). Previous approaches required a specialized combination of computer vision tools to answer specific questions about an image. By comparison, VLMs are designed to answer any question about an image. Questions are specified in natural language and are mostly answered accurately if the input is within its training distribution.

In this work, we introduce Semantic Symbolic State Estimation (S3E), the first zero-shot state estimator based on these VLMs. Our objective is to provide a general, versatile, and performant solution for state estimation that will accelerate the process of constructing state estimation functions for researchers and practitioners of task planning. S3E takes advantage of the foundation model's internal world model and semantic understanding (Schneider et al., 2024; Smeaton, 2024) to assess the likelihood of certain symbolic components of the environment's state that are relevant to the task being solved. It consists of two stages: (1) translating symbolic predicate definitions into natural language questions and (2) answering them given visual input. We show that the translation stage significantly improves performance in Appendix B. Fig. 1 demonstrates the usage of S3E in a real-world robotics task. We show that a zero-shot solution is indeed possible in some cases. To further improve performance alongside the VLM's strong priors, we use natural language instruction and targeted environment design to remove ambiguities and reduce uncertainties about the environment or task.

We analyze S3E as a multi-label classifier where the labels are the set of grounded predicates that make up all possible facts about the task state. Our experiments focus on usability and showcase a simulated and real-world robotic domain. S3E also achieves high performance in a photorealistic blocksworld with ever-changing objects in Appendix C, showing that it is truly general-purpose and versatile. We propose task-specific solutions to handle two kinds of uncertainties in our proposed state estimator. The first is the model's uncertainty regarding the state. The second stems from the subjective nature of the actual state relative to the intent of the task designer, i.e., whether a certain property holds for a given state is in the eye of the beholder. We show examples of these uncertainties and how they can be reduced using natural language instruction and minimal environment design. This improves on previous work that elicit uncertainties in language models Ren et al. (2023); Xiong et al. (2023) by leveraging this idea for symbolic state estimation in the context of task planning. Regardless of these uncertainties, general-purpose state estimation is a needed change from the specialized solutions offered by today's state-of-the-art.

This paper presents the following contributions:

- Introduction of Semantic Symbolic State Estimation (S3E): first zero-shot symbolic state estimator using vision-based foundation models.
- Proposal of a general solution for high-level state estimation in task planning.
- Identification and mitigation of model uncertainty and task-specific ambiguity.
- Empirical demonstration of S3E's effectiveness in simulated and real-world environments.

## 2 RELATED WORK

State estimation is vital for automated agents performing complex tasks, especially in Task and Motion Planning (TMP), where recognizing the state amidst uncertain dynamics is crucial for integrating high-level actions with physical motions (Curtis et al., 2024; Garrett et al., 2021; La-

griffoul et al., 2018; Kaelbling & Lozano-Pérez, 2011). Traditionally, state estimators have been hand-crafted (Moreno et al., 2024; Garrett et al., 2020), and modern task planning toolkits still rely on such manual coding (Wertheim et al., 2024). Learning-based methods, like Pankert & Hutter (2023), address specific tasks but lack generalizability and require exploration steps. In contrast, our approach eliminates manual coding and exploration, leveraging Vision-Language Models (VLMs) for zero-shot generalization.

Large Language Models (LLMs) have been explored in task planning (Kambhampati et al., 2024; Liu et al., 2023a; Huang et al., 2022) but typically overlook environment dynamics and sensing. Chen et al. (2024b) automate TMP using LLMs but depend on textual domain representations and ignore state uncertainty. Unlike these, S3E incorporates sensor input for state estimation.

Open-world planning has introduced LLMs in tasks where the agent lacks prior knowledge of its environment. Singh et al. (2023) assume known initial states and action effects, while Ding et al. (2023) predict planning obstacles without full state estimation. VLM-based state estimation approaches, such as Chen et al. (2024a), rely on external action success indicators. Duan et al. (2024b) uses VLMs in a robot manipulation pipeline that includes state estimation. However, the method requires a scene representation with task-specific elements and demonstrations while being coupled into a manipulation-centric system. Duan et al. (2024a) and Liu et al. (2023c) also use VLMs for scene understanding but focus on reasoning over action failure rather than state estimation.

Another string of related work is that of scene graph generation. These methods are designed to construct a matching graph of semantic entities and their relationships from a given scene representation. The scene representation can be 3D, e.g., mesh (Armeni et al., 2019) or point cloud Wang et al. (2023b) (which may be hard to come by), while others use images Zhao et al. (2023); Wu et al. (2023). Some approaches are even specifically geared toward perception and planning Maggio et al. (2024); Gu et al. (2024). Unlike these approaches, S3E requires no additional training beyond that of the pre-trained VLM. It does not need task-specific adaptation, making it more versatile out-of-the-box. Furthermore, S3E and scene graphs do not cancel each other out, i.e., they can be used together to further improve state estimation accuracy. For example, scene graph data may be used as additional input for S3E to mitigate uncertainties about the scene.

## 3 BACKGROUND

**Symbolic Task Domains**  In this work, we support agents with tasks for which the *task domain* (Geffner & Bonet, 2013) is defined as a tuple $\Sigma = \langle S, s_0, S_G, A, T, c \rangle$. $s_0 \in S$ is the initial state, $S_G \subseteq S$ is a set of goal states, $A(s) \subseteq A$ is the set of actions applicable at state $s$, $T$ is a deterministic transition function where $T(a, s) \in S$ is the state that follows $s$ after performing action $a$, and $c(a, s)$ is a positive cost of performing action $a$ at state $s$. A *task plan* is a sequence of actions $(a_1, ..., a_n)$ that are applicable in order from $s_0$ onwards and ultimately reach some state $s \in S_G$.

We focus on symbolic representations of task domains. Namely, $s$ is defined over a finite feature space $F$. Thus, each state $s \in S$ is an assignment over the set of features, i.e., $s = (f_1, f_2, ..., f_{|F|})$. This representation makes it easy to support representations such as Planning Domain Definition Language (PDDL) McDermott et al. (1998) which define actions using three sets of propositions over the features denoting the preconditions, add effects, and delete effects for that action. In Fig. 1, the pick-up action has 'robot-gripper-empty()' as a precondition, and 'robot-gripper-empty()' and 'in-table-section(spray-bottle, blue)' as delete effects. This highlights the important role state estimation has on the ability of automated agents to assess what is achievable in the current state.

**Instruction Tuned Vision-Language Models**  A Vision-Language Model (VLM) is a machine-learning model that combines natural language processing and computer vision (Li et al., 2022; Radford et al., 2021; Li et al., 2019) to perform various tasks. Visual Question Answering (VQA) is a task that uses VLMs to answer natural language questions about visual input, making them highly versatile question-answering functions.

In the general case, a VQA model is a parameterized function $g_\phi$ that accepts an image $X_v$ and text $X_q$ as input and outputs a probability distribution over a vocabulary of predefined tokens $V$. The input text is a sequence of tokens $X_q = (X_q^1, ..., X_q^n)$ in $V$, with sequence length $n$. The output sequence $X_a = (X_a^1 + ... + X_a^m)$ of length $m$ is generated by sampling tokens in an autoregressive

fashion. First, the token $X_a^1 \sim g_\phi(X_v, X_q)$ is sampled. The next token is then sampled with a concatenation of the same text and the new token $X_a^2 \sim g_\phi(X_v, X_q + X_a^1)$, and repeating this process sequentially until an "end-of-response" token is generated $X_a^m \sim g_\phi(X_v, X_a + X_a^1 + ... + X_a^{m-1})$. The final model's response is the sequence of all sampled tokens $X_a = (X_a^1 + ... + X_a^m)$. VQA models are trained such that $X_q$ is a question about $X_v$ and $X_a$ is likely the correct answer to the question. For more information on instruction-tuning these models, see Appendix A.

## 4 USING SEMANTICS FOR STATE ESTIMATION

Our objective is to provide a general and versatile solution for state estimation that will accelerate the construction of state estimation functions for researchers and practitioners of task planning. We want to provide a function that, given a symbolic world model, estimates the individual state features.

Let $\Phi$ be the set of possible agent image observations in a fully observable setting. Given task domain $\Sigma = \langle S, s_0, S_G, A, T, c \rangle$ where $S$ is defined over features $F$, let $\xi_\Sigma$ be the ground truth state estimator, that is, $\xi_\Sigma(X_v) = (f_1, .., f_{|F|}) \in S$ is the true feature assignment corresponding to observation $X_v \in \Phi$. We want to find meta function $\xi_G$ that accepts a task domain as input and outputs an approximate state estimator function, i.e., $\forall X_v \in \Phi, \ \xi_G(\Sigma)(X_v) \approx \xi_\Sigma(X_v)$. In simple terms, we want to find a global function $\xi_G$ that outputs a task-specific state estimator function $\xi_G(\Sigma)$ that reliably approximates ground truth estimator $\xi_\Sigma$.

A naïve approach to defining $\xi_G(\Sigma)$ is to learn from applying actions from different states and pairing the resulting observations with the expected state according to the domain description. But if we could fully trust actions and the controller executing their trajectory, the *downward refinement property* (Bacchus & Yang, 1991) would hold, meaning low-level actions would guarantee the desired task state. Thus, the agent can blindly execute planned actions sequentially and reach the goal with certainty, making the state estimator redundant. Since this assumption is unrealistic, a different approach is needed. In this work, we turn to semantics as the source from which the state is derived.

In real-world symbolic task planning, it is common for the task designer to maintain states and actions that contain some semantic value. In Example 1, the "pick-up(item)" action has a clear semantic meaning: the item is picked up by the agent. Therefore, we assume that the task domain components can be clearly described using natural language. We use these descriptions as a semantic guideline for state estimation.

We present Semantic Symbolic State Estimation (S3E), a semantic approach to state estimation that uses pre-trained vision-based foundation models to provide a general and versatile solution that generates a joint probability distribution over the symbolic state components. This way, we take advantage of one of the great strengths of foundation models: their internal world model and semantic understanding (Schneider et al., 2024; Smeaton, 2024). Specifically, we translate a textual domain description into a collection of natural language queries for which answers determine the features of a state using a Large Language Model (LLM). We then answer these questions in reference to vision-based observations, denoted $X_v$, using a VQA model, thus estimating the current task state.

Using a VQA model as a state estimator requires making some assumptions: (1) the image observations contain the information required to determine the task state (e.g., no object occlusion); (2) the domain description is unambiguous (e.g., use object names like "white-table" and "black-table" instead of "table1" and "table2"); (3) all objects are visually distinguishable (no identical objects). Note that while assumption 3 seems particularly demanding, it can easily be overcome by labeling objects using a combination of object detection and object tracking methods, e.g., YOLO (Redmon et al., 2016) and SORT (Bewley et al., 2016). Additionally, S3E assumes that the state space features are defined as semantic predicates that refer to one or more objects.

A predicate $P$ is a function that that represents a property of relation between compatible object parameters $\omega = (\omega_1, ..., \omega_m)$. Denote $\Omega_P$ the set of object sequences that are valid arguments for predicate $P$. A grounded predicate is a predicate-parameters pair $P(\omega)$ where $\omega \in \Omega_P$. The feature space $F$ corresponding to state space $S$ is defined as the set of all grounded predicates. For each grounded predicate, we would like to answer the question "In this image, does $P(\omega)$ hold true?". However, we would like to ask these questions in natural language, e.g., if $P$ is "on-table" and $\Omega$ is "milk-carton, wood-table", then we would like to answer the question "In this image, is the milk carton on the wooden table?". We perform this translation from grounded predicate to natural

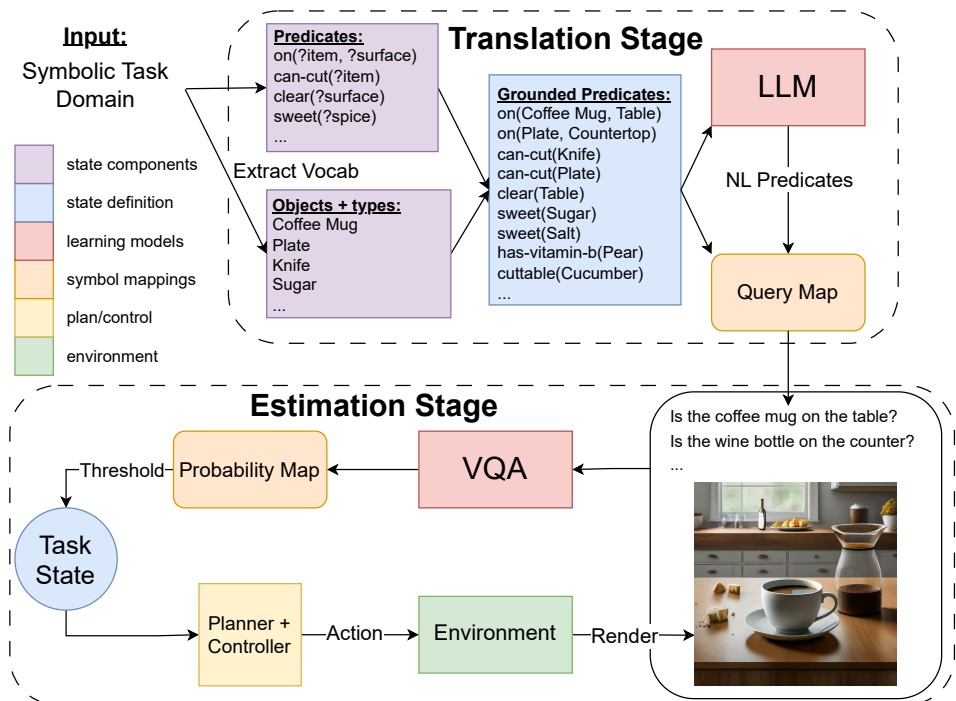

Figure 2: The S3E pipeline. **Translation State (top):** A set of grounded predicates that determine the task state is extracted from a symbolic task domain. Grounded predicates are translated into natural language queries. **Estimation Stage (bottom):** A VLM performs VQA on a rendering of the environment and each predicate question individually. Predicates are mapped to truth probabilities. Highly likely predicates are considered part of the task state and are provided to the planner.

language question using a LLM. We expect the model to answer "yes" or "no" according to what a human would most likely say, making this task adequate for a VQA model. In this work, we assume that predicates are boolean functions. However, this can easily be enhanced to numeric functions by retrieving numeric answers from the VQA model rather than boolean answers.

Fig. 2 depicts the S3E pipeline. It is divided into two stages: the *translation stage* and the *estimation stage*. The translation stage creates a mapping from grounded predicates to natural language questions. This is done once for a given set of predicates and objects and can be reused for different goal states. The questions are then answered repeatedly at each state during the estimation stage.

**Translation Stage.** Initially, we extract a finite vocabulary of predicates of arbitrary size $n$, denoted $\{P_i\}_{i=1}^n$, and objects of interest from task domain $\Sigma$. We combine predicates and compatible sequences of object parameters into grounded predicates $\bigcup_{i=1}^n \{P_i(\omega) | \omega \in \Omega_{P_i}\}$. A truth assignment of each grounded predicate defines a high-level task state.

We then translate grounded predicates into natural language queries using a LLM to directly translate the predicate name to a natural language question. We use these to construct a queries map $Q$, i.e., a mapping from grounded predicates to their natural language counterparts $Q(P_i(\omega)) = X_{q_{P_i(\omega)}}$. This step is important since instruction-tuned VQA models are trained to answer questions, not to determine the truth values of predicates. Using models as they were trained allows us to use a smaller model for higher efficiency in both speed and compute power. An ablation study of the translation stage is shown in Appendix B.

**Estimation Stage.** Given an image (or several images) $X_v$ rendered from the task environment, we invoke the VQA model with the questions corresponding to each predicate separately. The model is instructed with instruction text $X_I$ to answer only with "yes" and "no" as these are the values of interest in binary state estimation. For grounded predicate $P_i(\omega)$, this yields probability distribution $g_\phi(X_v, X_I + X_{q_{P_i(\omega)}})$. From this distribution we extract the probabilities of "yes" and

"no" tokens (multiple tokens for each) and calculate the normalized probability of "yes" vs. "no", denoted $Y_{P_i(\omega)}$. This induces a mapping between grounded predicates and a truth probability.

Once predictions are made for all grounded predicates, we threshold their probabilities to obtain binary values for each one. Specifically, for a given observation $X_v$, the S3E state estimation is defined $\xi_G(\Sigma)(X_v) = \left(Y_{P_i(\omega)} \geq \theta \middle| 1 \leq i \leq n, \omega \in \Omega_{P_i}\right)$ for some threshold $\theta \in [0, 1]$. The estimated state is monitored by a planner/controller unit which chooses the next action to perform. After action execution, the environment is rendered again and the estimation process is reiterated.

## 5 UNCERTAINTY IN SEMANTIC STATE ESTIMATION

When employing vision-based semantic state estimation, as outlined in Fig. 2 and under the assumptions of Section 4, two distinct kinds of uncertainties emerge: model uncertainty and task uncertainty.

Model uncertainty pertains to the system's lack of knowledge about the world. For instance, if the training data does not include a cereal box (neither in images nor text), the state estimation, as demonstrated in Example 1, would likely result in random guesses for any predicate involving the "cereal" object. This type of uncertainty reflects the model's limited exposure during training and its inability to generalize beyond its learned domain.

Task uncertainty, on the other hand, arises from the subjective nature of task design and interpretation. A pertinent example can be seen in Fig. 2, where it may be ambiguous whether a coffee mug is "on the table" if a small plate is separating the two. This uncertainty stems from differing perspectives or intentions in defining predicates.

We classify these uncertainties based on their origins: aleatoric or epistemic (Hüllermeier & Waegeman, 2021). Model uncertainty can be viewed as aleatoric because it involves inherent randomness in the world, yet it can also be epistemic since expanding the training dataset could mitigate it. Task uncertainty, similarly, can be aleatoric due to variability in task design but may also be epistemic, as clearer feedback or instructions from the task designer can reduce it. For clarity, we adopt the following conventions: (a) **Model uncertainty** is treated as aleatoric, with the model parameters fixed. Our mitigation strategy focuses on designing experiments to avoid ambiguous or unclear states entirely. (b) **Task uncertainty** is considered epistemic, and we address it by providing explicit, clarifying instructions to refine predicate definitions and reduce ambiguity.

The uncertainties faced by S3E are akin to those encountered by humans during manual state estimation. To manage these uncertainties effectively, we implement two strategies inspired by human practices: (1) **Few-shot adaptation**: In our simulated pick-and-place experiment, we provide examples to guide the state estimator, such as describing specific object appearances; (2) **Environmental and action design**: We structure the environment and robot behavior to make states unambiguous. For instance, in our experiments, the robot assumes a standardized configuration after each action, ensuring clarity on whether it is holding an object. These approaches, detailed and demonstrated in Section 6, serve to minimize both types of uncertainties and enhance the robustness of our state estimation framework.

## 6 EMPIRICAL EVALUATION

Our empirical evaluation aims to demonstrate that S3E can estimate the high-level state with minimal task-specific enhancements. We demonstrate this in a grocery sorting pick-and-place setting with natural language instructions and deliberate domain setup that clarifies certain aspects of the environment for the VLM. We analyze both a simulated and a real-world example. All code will be made public upon acceptance of this paper.

While our primary emphasis is on usability, we also showcase the adaptability of S3E in a photorealistic block world environment (Asai, 2018). This environment features a wide variety of objects that change between tasks. Not only is blocksworld a well-studied and challenging problem in task planning, but also photorealistic blocksworld is based on, CLEVR, a common dataset for evaluating neuro-symbolic understanding Mao et al. (2022); Johnson et al. (2017). Experiment details and results for this domain can be found in Appendix C.

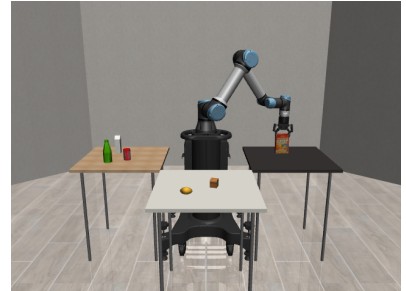

Figure 3: Example renderings from our simulated grocery sorting domain.

Figure 4: An example state where it is unclear whether the object is gripped.

**Experiment 1: Simulated Grocery Sorting**   This experiment analyzes our state estimation pipeline from Fig. 2 as a multilabel classifier of the task predicates. We attempt to estimate the state of an environment designed for sorting groceries onto different tables as in Example 1. We use a simulated environment with a robotic arm and semi-realistic objects from the robosuite framework (Zhu et al., 2022). It contains 3 tables (wood, black, and white) and 6 grocery items (milk, bread, lemon, can, cereal, and bottle). Fig. 3 shows different rendering viewpoints of this domain.

The agent can perform "pick" and "place" actions of specific objects from and onto a specified table. The task is defined in PDDL (McDermott et al., 1998), a common description language for task domains, which requires explicitly declaring predicates and objects of interest. These are extracted using an off-the-shelf parser (Micheli & Bit-Monnot, 2022) and fed into the S3E pipeline.

The advantage of running in simulation is that we can collect data more easily. The simulation can run faster than a real-world equivalent, and constructing a ground-truth state estimator is easier with access to privileged simulator information. We collect pairs of rendered and task states by randomly setting the items' positions on the tables and having the robotic agent perform random actions that perturb the environment and produce a new rendered state. We collect 2,000 data points[1] using the procedure described in Appendix D. The agent performs the actions in full instead of us directly placing the items on the tables or in the robot's gripper to achieve natural-looking states that may be reached when solving the task, e.g., items knocked over, some on the floor, various grip positions, etc. Actions are implemented imperfectly to achieve these real-world situations.

Each data point is processed through the pipeline depicted in Fig. 2. The quality of estimation will be determined using evaluation tools for multilabel classification. We expect to witness two kinds of failures (see Section 5). First, how the actions are designed sometimes makes it unclear whether the robot is gripping an object. This is an example of aleatoric uncertainty relative to the scene and the underlying task. In an attempt to overcome this, we run the same experiment on an alternative dataset wherein we assert a certain robot "home pose" after picking up and placing an object. The home pose is the pose of the robot as shown in Fig. 3, where it can be clearly seen whether or not an object is within the gripper jaws. These experiments are denoted with "Pose". Second, because this is a simulated environment and VQA systems are normally trained on real-world data, we fully expect low performance on predicates that consider the less realistic items. This will illustrate the uncertainty introduced by the limitation of the model's training distribution. To overcome this, we prompt the VLM with additional information about the task and its objects, e.g., "The milk carton is a clean white rectangular box with a triangular top". We denote these experiments with the "Instruct" label. The exact instruction prompt used in these experiments is provided in Appendix E.3.

**Experiment 2: Real-world Grocery Sorting**   The purpose of this experiment is to demonstrate the real-world applicability of our method and to address the VLM's difficulty dealing with simulated data. Similarly to the previous experiment, a robot arm must move items between different sections on a table. The items in the scene are a green mug, a water bottle, a soda can, and a window cleaner spray bottle. The table is divided into a white section and a blue section. A single camera

---

[1]In our testing, this was a large enough sample for a diverse and representative dataset.

is pointed at the scene. Fig. 1a shows an image captured in this environment. Like before, the robot can perform the same pick and place actions, and the task is defined using PDDL.

Unlike the simulated environment in experiment 1, collecting thousands of data points is unrealistic, and implementing a ground-truth state estimator is not straightforward. Instead, we have the agent solve a simple rearrangement task and estimate the state at each frame of a captured video. We then manually check the results for each frame and measure approximate performance for S3E. We also separate the middle frames in between actions, denoted "Mid-poses", to measure the performance for frames in which the agent must choose its next action or determine that the goal has been achieved. The pick-and-place actions were implemented to avoid object collisions and occlusions.

## 6.1 SETUP

To implement the flow in Fig. 2, we require a LLM for the translation stage and VQA model for the estimation stage. We use Large Language model Meta AI (LLaMA) 3 (Touvron et al., 2023) as the translator, and Large Language and Vision Assistant (LLaVA) (Liu et al., 2023b), specifically the OneVision (OV) model (Li et al., 2024), as the instruction-tuned VQA model. We chose these models because they are leaders in the open-source world, making them freely available for research, unlike proprietary models. Furthermore, we chose to use OV for its outstanding performance in VLM benchmarks and because it was trained on 3D data using multi-viewpoint images. However, we do not use multi-image inputs as this uses significantly more GPU memory. This is critical for robotics applications where the agent must carry its computing power onboard.

We compare three OV model sizes: 0.5B, 7B, and 72B, where XB denotes the model's size in billions of parameters. The concrete prompts used to instruct the models to perform the state estimation can be found in Appendix E. The exact hardware specifications can be found in Appendix F.

For simulation, we use the MuJoCo physics engine (Todorov et al., 2012). The robots used in experiments 1 and 2 are the UR5 and UR5e, respectively. Task-defined actions used for data collection are implemented using privileged information (the objects' locations and dimensions) to infer the desired grasp poses. Motions between poses are planned using RRT* (Karaman & Frazzoli, 2011).

## 6.2 RESULTS

**Experiment 1: Simulated Environment** This section evaluates the S3E state estimators in Experiment 1, treating them as multilabel classifiers. Since most predicates (approximately 75%) are false (e.g., when one object is being gripped, others are not), a baseline accuracy of around 75% can be achieved by predicting all predicates as false. Therefore, we focus on the Average Precision (AP) (Average Precision) score with micro and macro averaging as our primary metrics (Sokolova & Lapalme, 2009).

Table 1 presents accuracy and AP scores for different S3E VLM models. As anticipated, larger models show better performance. The 0.5B model performs poorly, with an AP score below 50%. The 7B model achieves AP scores ranging from 66% to 77%, while the 72B model scores between 74% and 91%. The "Pose" modification significantly improves performance across all models, particularly for the macro average, which emphasizes the gripping predicates: improvements are approximately 21% for the 0.5B model, 16% for the 7B model, and 12% for the 72B model.

Adding natural language instructions negatively impacts the 0.5B and 7B models, likely due to confusion from the additional context. In contrast, the 72B model shows consistent AP performance for micro averages (about a 1% difference) and substantial macro average gains (approximately 4% without "Pose" and 9% with it). Combining both enhancements ("Instruct + Pose"), the 72B model improves by approximately 9.5% (micro) and 22% (macro). Similar trends are observed in the photorealistic blocksworld domain (see Appendix C).

Although accuracy alone offers limited insight, examining it across different thresholds reveals model certainty. The 72B model maintains balanced certainty around the 50% mark, effectively distinguishing between true and false predicates. In contrast, the 0.5B and 7B models show increased accuracy at higher thresholds, suggesting they often assign high probabilities to false values. Setting a higher threshold for these models may reduce false positives.

Table 1: A comparison of tested S3E VLM instances in experiment 1 (simulated) on accuracy (3 thresholds $\theta$) and AP scores.

|  | $\theta = 0.3$ | $\theta = 0.5$ | $\theta = 0.7$ | AP (micro) | AP (macro) |
|---|---|---|---|---|---|
| 0.5B | 0.78 | 0.79 | 0.79 | 0.37 | 0.35 |
| 7B | 0.72 | 0.85 | 0.88 | 0.70 | 0.66 |
| 72B | 0.82 | 0.90 | 0.89 | 0.81 | 0.74 |
| 0.5B + Instruct | 0.53 | 0.71 | 0.76 | 0.24 | 0.33 |
| 7B + Instruct | 0.78 | 0.85 | 0.85 | 0.67 | 0.63 |
| 72B + Instruct | 0.88 | 0.92 | 0.89 | 0.80 | 0.78 |
| 0.5B + Pose | 0.78 | 0.78 | 0.78 | 0.38 | 0.42 |
| 7B + Pose | 0.74 | 0.87 | 0.90 | 0.76 | 0.77 |
| 72B + Pose | 0.86 | 0.93 | **0.91** | 0.87 | 0.83 |
| 0.5B + Instruct + Pose | 0.53 | 0.71 | 0.75 | 0.24 | 0.41 |
| 7B + Instruct + Pose | 0.81 | 0.86 | 0.86 | 0.73 | 0.73 |
| 72B + Instruct + Pose | **0.90** | **0.94** | **0.91** | **0.88** | **0.91** |

Table 2: A comparison of tested S3E VLM instances in experiment 2 (real-world) on accuracy (3 thresholds $\theta$) and AP scores.

|  | $\theta = 0.3$ | $\theta = 0.5$ | $\theta = 0.7$ | AP (micro) | AP (macro) |
|---|---|---|---|---|---|
| 0.5B | 0.58 | 0.63 | 0.67 | 0.45 | 0.61 |
| 7B | 0.78 | 0.77 | 0.76 | 0.79 | 0.84 |
| 72B | 0.81 | **0.81** | **0.82** | 0.86 | 0.91 |
| 0.5B Mid-poses | 0.56 | 0.63 | 0.66 | 0.46 | 0.74 |
| 7B Mid-poses | 0.73 | 0.77 | 0.77 | 0.80 | 0.85 |
| 72B Mid-poses | **0.82** | 0.81 | 0.81 | **0.90** | **0.99** |

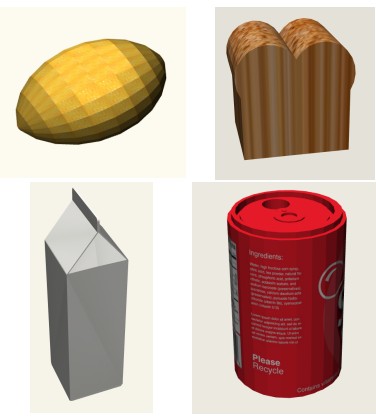
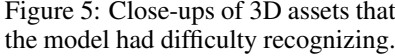

Figure 5: Close-ups of 3D assets that the model had difficulty recognizing.

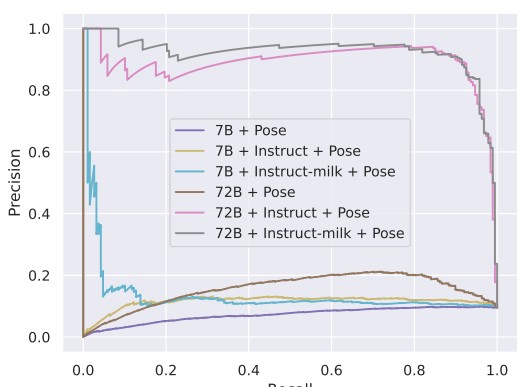

Figure 6: Precision-recall curves for the "robot-gripping(milk-carton)" predicate.

Detailed AP scores per predicate (see Fig. 9 in Appendix G) highlight two key observations. First, some objects are more challenging for the model to identify (Fig. 5). Second, distinguishing whether an object is gripped can be unclear. For instance, Fig. 4 shows ambiguity in the robot's grip on a cereal box. Fig. 7 shows AP scores for grip-related predicates, with notable improvements when task-specific modifications are applied. However, objects like bread, milk cartons, and soda cans remain difficult for the model to recognize.

The most notable improvement occurs with the "Instruct" modification for the 72B model (see pink bars in Fig. 7). Instructions negatively impact the 0.5B and 7B models, so these results are omitted. Fig. 6 compares precision-recall curves for the 7B and 72B models, showing the value of natural language instructions for the "robot-gripping(milk-carton)" predicate. Further improvements are possible with predicate-specific instructions (e.g., "instruct-milk"). However, even with enhanced instructions, the bread object remains unrecognized, likely due to its unrealistic 3D model (Fig. 5, top right). This highlights limitations in current VLMs compared to human perception.

**Experiment 2: Real-World Robot** Similar to Experiment 1, we use the AP score as the primary metric since most predicates are usually false. With fewer predicates, accuracy is lower if all are predicted false. Table 2 displays results for different VLM models. The standalone models (0.5B, 7B, 72B) predict the state at each frame, while "Mid-poses" indicate predictions between actions. A sample video with annotations is available in the supplementary material.

Performance on real-world images is stronger than in the simulation, reflecting the models' training data. Improvements are notable: 0.5B improves by ~74.5%, 7B by ~27.5%, and 72B by ~22%. "Mid-poses" show even greater gains: ~110.5%, ~29.5%, and ~32%, respectively. The 72B model achieves near-perfect estimation (AP >99%) in "Mid-poses," though accuracy indicates uncertainty. This underscores the value of environment-specific descriptions for reducing ambiguity.

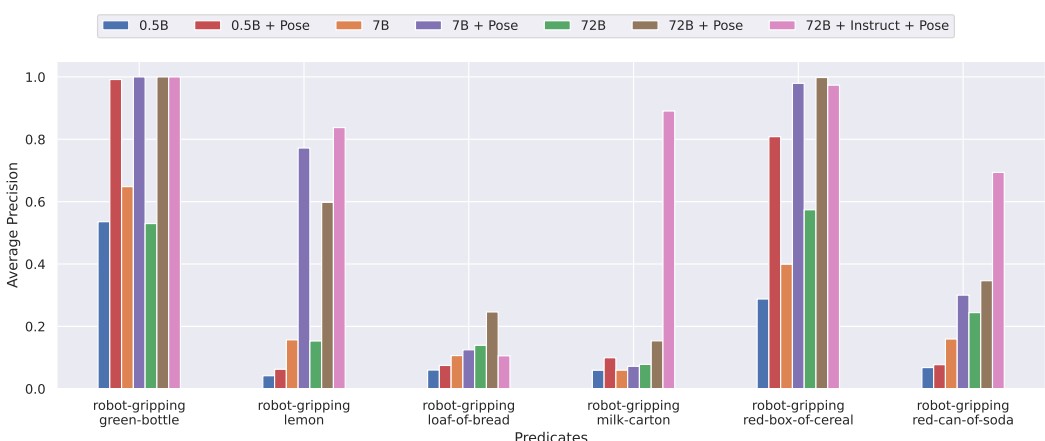

Figure 7: AP scores comparison for all object-gripping predicates with and without task-specific modifications.

## 6.3 DISCUSSION

Our experiments demonstrate that S3E can approximate symbolic states with over 90% precision in both real and simulated tasks. We also see similar results for our additional experiments that test our method's adaptability (see Appendix C). This high accuracy showcases the potential of our novel pipeline to estimate symbolic states in task planning environments. By harnessing the VQA model's semantic understanding, we bridge the gap between visual inputs and symbolic state representations.

While S3E struggles with out-of-distribution scenarios, targeted environment manipulation and task-specific natural language instructions help mitigate this issue, particularly in larger models. Notably, S3E combined with the OV model performs better in real-world environments than simulated ones, likely due to the composition of its training data. These findings highlight both the strengths of our approach and areas for future refinement in VQA, particularly for state estimation.

## 7 CONCLUSION

In this work, we presented S3E, a general-purpose vision-based symbolic state estimator using VLMs. S3E offers a versatile replacement for hand-crafted state estimator functions that are specialized for the individual task. While our framework was used deterministically, it can easily be adapted for probabilistic estimations and account for belief state updates in partially observable domains Kaelbling et al. (1998). We intend to explore this in future work.

We empirically evaluated S3E coupled with LLaMA 3 and LLaVA OV as a multiclass classifier of task-specific predicates for robot pick-and-place tasks. We showed that this combination can achieve over 90% state estimation precision with no task-specific coding involved. In a simulated environment, we demonstrated two kinds of uncertainties brought on by using a VQA model for state estimation. We showed how to reduce these uncertainties with low-effort modifications to the environment and natural language instructions to the model. In a real-world setting, we showed that high performance can be achieved without providing any task-specific information.

While S3E offers a general solution to visual state estimation, it comes with some limitations relative to hand-crafted state estimators. It requires a visual input setup with full observability. Furthermore, it requires that all objects be visually distinguishable. We also perform an exhaustive search over all grounded predicates, which can become computationally expensive in more complex environments. Finally, the task must be defined in descriptive language to generate high-quality queries for the VQA model. Future work should address the above limitations, uncertainty detection (e.g. using confidence scores, discrepancy analysis, or external knowledge integration) and further mitigation (e.g. Bayesian state estimation, conformal prediction), and improve performance with task-specific information (e.g., using predicate correlations).

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

## A   INSTRUCTION-TUNED MODELS

LLM instruction-tuning is a method for improving a model's ability to follow natural language instructions (Wang et al., 2023a; Ouyang et al., 2022). To achieve this, models are fine-tuned using enhanced datasets of instruction-output pairs. These datasets often include question-answer pairs, task-completion examples, dialogue simulations, solved coding exercises, and more. Training is typically performed using supervised learning. A reward model is learned through supervised output and the language model is trained to maximize reward. To ensure human compatibility, this process involves human-curated datasets, annotations, and response rankings (Ouyang et al., 2022), but this is also done using another LLM for scalability (Wang et al., 2023a).

A cross-breed of VLMs and instruction-tuning yielded LLaVA (Liu et al., 2023b), a VLM trained to follow natural-language instructions based on visual input. This allows giving the model a certain context in which to answer input questions. We take advantage of this ability to put the agent into the context of the task planning problem at hand.

Table 3: An ablation test of the translation stage for S3E VLM instances in experiment 1 (simulated) on accuracy (3 thresholds $\theta$) and AP scores. Entries labeled "(no trans)" do not include the translation stage.

|  | $\theta = 0.3$ | $\theta = 0.5$ | $\theta = 0.7$ | AP Score (micro) | AP Score (macro) |
|---|---|---|---|---|---|
| 0.5B | 0.78 | 0.79 | 0.79 | 0.37 | 0.35 |
| 0.5B (no trans) | 0.79 | 0.79 | 0.79 | 0.18 | 0.23 |
| 7B | 0.72 | 0.85 | 0.88 | 0.70 | 0.66 |
| 7B (no trans) | 0.57 | 0.63 | 0.70 | 0.26 | 0.50 |
| 72B | 0.82 | 0.90 | 0.89 | 0.81 | 0.74 |
| 72B (no trans) | 0.81 | 0.87 | 0.85 | 0.73 | 0.68 |
| 0.5B + Pose | 0.78 | 0.78 | 0.78 | 0.38 | 0.43 |
| 0.5B (no trans) + Pose | 0.78 | 0.78 | 0.78 | 0.19 | 0.25 |
| 7B + Pose | 0.74 | 0.87 | 0.90 | 0.76 | 0.77 |
| 7B (no trans) + Pose | 0.60 | 0.64 | 0.70 | 0.27 | 0.59 |
| 72B + Pose | 0.86 | 0.93 | 0.91 | 0.87 | 0.83 |
| 72B (no trans) + Pose | 0.84 | 0.89 | 0.87 | 0.80 | 0.75 |
| 0.5B + Instruct | 0.53 | 0.71 | 0.76 | 0.24 | 0.33 |
| 0.5B (no trans) + Instruct | 0.71 | 0.79 | 0.79 | 0.17 | 0.23 |
| 7B + Instruct | 0.78 | 0.85 | 0.85 | 0.67 | 0.63 |
| 7B (no trans) + Instruct | 0.53 | 0.58 | 0.61 | 0.21 | 0.47 |
| 72B + Instruct | 0.88 | 0.92 | 0.89 | 0.80 | 0.78 |
| 72B (no trans) + Instruct | 0.81 | 0.87 | 0.85 | 0.73 | 0.68 |
| 0.5B + Instruct + Pose | 0.53 | 0.71 | 0.75 | 0.24 | 0.42 |
| 0.5B (no trans) + Instruct + Pose | 0.71 | 0.78 | 0.78 | 0.18 | 0.25 |
| 7B + Instruct + Pose | 0.81 | 0.86 | 0.86 | 0.73 | 0.73 |
| 7B (no trans) + Instruct + Pose | 0.54 | 0.59 | 0.61 | 0.22 | 0.55 |
| 72B + Instruct + Pose | 0.90 | 0.94 | 0.91 | 0.88 | 0.91 |
| 72B (no trans) + Instruct + Pose | 0.84 | 0.89 | 0.87 | 0.80 | 0.75 |

## B   TRANSLATION STAGE ABLATION

The goal of this experiment is to emphasize the importance of the translation stage of S3E (see Section 4). To do this, we instruct the VLM to answer "true" or "false" given a predicate and extract the probability for that predicate as in the S3E estimation stage. We show this for both experiments defined in Section 6.

We used the following system prompt for the VLM:
*The following is a PDDL domain*
*{DOMAIN}*
*Here are the names of all the objects in the current problem, sorted by their type:*
*{OBJECTS_BY_TYPE}*
*Given a grounded predicate with concrete variables, state whether the statement is true or false.*
*Respond only with a "true" or "false" response and nothing else.*
where *DOMAIN* and *OBJECTS_BY_TYPE* are as defined in Appendix E.1. Given a grounded predicate, the VLM is prompted with the predicate followed by its variables, comma separated in parentheses (e.g. "on-table(lemon,black-table)").

The results in Tables 3 and 4 clearly show that the translation stage improves AP score for all S3E VLM instances for both macro and micro averaging. This is especially true for the smaller models (0.5B and 7B), seeing improvements of over 200% in micro AP and over 50% in macro AP. The vast improvement in micro AP points to an issue in understanding the most common predicates when the translation stage is skipped. We conclude that the translation stage is a necessary step to enable the VLM to understand the questions that each predicate poses and what it means in the scene.

The performance improvement when applying the translation stage is also visible for the accuracy metric in most cases. But due to class imbalance, The accuracy for a given threshold may not be

Table 4: An ablation test of the translation stage for S3E VLM instances in experiment 2 (real-world) on accuracy (3 thresholds $\theta$) and AP scores.

|  | $\theta = 0.3$ | $\theta = 0.5$ | $\theta = 0.7$ | AP Score (micro) | AP Score (macro) |
|---|---|---|---|---|---|
| 0.5B | 0.58 | 0.63 | 0.67 | 0.46 | 0.55 |
| 0.5B (no trans) | 0.47 | 0.60 | 0.64 | 0.40 | 0.48 |
| 7B | 0.78 | 0.77 | 0.76 | 0.79 | 0.81 |
| 7B (no trans) | 0.44 | 0.49 | 0.54 | 0.36 | 0.45 |
| 72B | 0.81 | 0.81 | 0.82 | 0.86 | 0.89 |
| 72B (no trans) | 0.66 | 0.70 | 0.72 | 0.78 | 0.75 |
| 0.5B Mid-poses | 0.56 | 0.63 | 0.66 | 0.47 | 0.70 |
| 0.5B (no trans) Mid-poses | 0.51 | 0.58 | 0.66 | 0.43 | 0.59 |
| 7B Mid-poses | 0.73 | 0.77 | 0.77 | 0.80 | 0.83 |
| 7B (no trans) Mid-poses | 0.46 | 0.53 | 0.56 | 0.40 | 0.60 |
| 72B Mid-poses | 0.82 | 0.81 | 0.81 | 0.90 | 0.99 |
| 72B (no trans) Mid-poses | 0.64 | 0.69 | 0.73 | 0.82 | 0.85 |

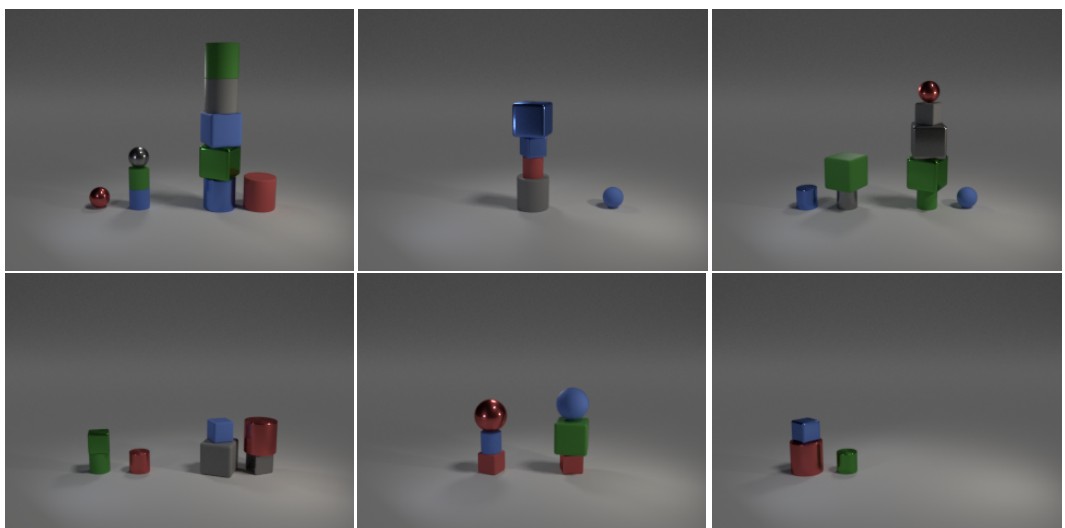

Figure 8: Renderings of random samples from the photorealistic blocksworld domain.

informative by itself. Surprisingly, The same pattern emerges as in the original experiments. That is, without translation, the certainty of the model is balanced around 50% for experiment 1, but a larger confidence threshold is needed for experiment 2.

## C  OBJECT DIVERSITY EXPERIMENT WITH PHOTOREALISTIC BLOCKSWORLD

The goal of this experiment is to showcase the adaptability of S3E in the face of vast object diversity. Here, we use a photorealistic version of the blocksworld domain Asai (2018). This domain contains objects that vary in size (small and large), color (8 different values), material (rubber and metal), and shape (cube, cylinder, and sphere). Objects can be on the table or stacked on top of each other. Given a state, a 3D scene is synthesized and rendered from a single viewpoint. Fig. 8 shows example renderings from this domain.

We collected over 7,500 data points in this domain using the same procedure as in experiment 1 (see Appendix D). Possible actions include moving a block from the table onto another block, from on

Table 5: A comparison of tested S3E VLM instances in photorealistic blocksworld with different limits on the number of objects. Compared metrics are accuracy (3 thresholds $\theta$) and AP scores.

| | $\theta = 0.3$ | $\theta = 0.5$ | $\theta = 0.7$ | AP Score (micro) | AP Score (macro) |
|---|---|---|---|---|---|
| 0.5B ($\leq 3$) | 0.72 | 0.81 | 0.81 | 0.44 | 0.76 |
| 0.5B ($\leq 5$) | 0.70 | 0.84 | 0.87 | 0.32 | 0.60 |
| 0.5B ($\leq 7$) | 0.72 | 0.86 | 0.89 | 0.26 | 0.45 |
| 0.5B ($\leq 10$) | 0.74 | 0.88 | 0.91 | 0.20 | 0.30 |
| 0.5B + Instruct ($\leq 3$) | 0.74 | 0.81 | 0.81 | 0.41 | 0.75 |
| 0.5B + Instruct ($\leq 5$) | 0.73 | 0.85 | 0.87 | 0.29 | 0.59 |
| 0.5B + Instruct ($\leq 7$) | 0.74 | 0.87 | 0.89 | 0.22 | 0.44 |
| 0.5B + Instruct ($\leq 10$) | 0.74 | 0.89 | 0.91 | 0.16 | 0.29 |
| 7B ($\leq 3$) | 0.82 | 0.84 | 0.85 | 0.68 | 0.91 |
| 7B ($\leq 5$) | 0.78 | 0.81 | 0.83 | 0.59 | 0.83 |
| 7B ($\leq 7$) | 0.75 | 0.78 | 0.82 | 0.50 | 0.71 |
| 7B ($\leq 10$) | 0.73 | 0.77 | 0.81 | 0.41 | 0.55 |
| 7B + Instruct ($\leq 3$) | 0.85 | 0.86 | 0.87 | 0.73 | 0.89 |
| 7B + Instruct ($\leq 5$) | 0.81 | 0.84 | 0.87 | 0.62 | 0.80 |
| 7B + Instruct ($\leq 7$) | 0.79 | 0.83 | 0.86 | 0.51 | 0.69 |
| 7B + Instruct ($\leq 10$) | 0.77 | 0.82 | 0.86 | 0.41 | 0.53 |
| 72B ($\leq 3$) | 0.92 | 0.92 | 0.93 | 0.88 | 0.95 |
| 72B ($\leq 5$) | 0.87 | 0.88 | 0.89 | 0.78 | 0.87 |
| 72B ($\leq 7$) | 0.84 | 0.86 | 0.87 | 0.68 | 0.76 |
| 72B ($\leq 10$) | 0.81 | 0.83 | 0.86 | 0.56 | 0.59 |
| 72B + Instruct ($\leq 3$) | 0.91 | 0.92 | 0.93 | 0.94 | 0.96 |
| 72B + Instruct ($\leq 5$) | 0.83 | 0.85 | 0.87 | 0.85 | 0.88 |
| 72B + Instruct ($\leq 7$) | 0.78 | 0.81 | 0.83 | 0.76 | 0.78 |
| 72B + Instruct ($\leq 10$) | 0.74 | 0.77 | 0.80 | 0.64 | 0.64 |

top of a block onto the table, or from on top of one block onto another. Only blocks with no other blocks stacked on top of them may be moved. Blocks cannot be stacked on top of cylinder types. Upon environment reset, a random number of objects $n$ is chosen between 2 and 10, and then $n$ unique objects are generated with a random size, color, material, and shape. This ensures that the dataset contains a diverse set of objects.

As in the main experiments, the class labels are severely imbalanced as predicates are usually false, making accuracy less informative. Additionally, since objects change between environment resets, many predicates are present only a few times throughout our collected dataset. Therefore, the difference between micro and macro averaging is expected to be much more extreme. Furthermore, macro averaging gains extra significance since it treats rare and frequent labels equally.

The number of objects in the scene have a significant effect on the performance of S3E Table 5 compares S3E performance on the photorealistic blocksworld with varying limits on the number of objects sampled We see steady AP improvements that range from ~61% in the largest model to over 150% in the smallest model when reducing from a 10 objects limit to a 3 objects limit.

One reason for the big difference in performance between object limits is that with more objects it is more likely that some objects are hard to differentiate. This can be seen in Table 6 where S3E performance is compared on different subsets of the dataset while keeping the 10 objects limit. We observe significant improvements in performance across the board compared to the full dataset with the 10 objects limit. Using S3E on a subset of a single material we see over 90% macro AP for both the 7B and 72B models. The highest improvement in performance is seen when disallowing colors to repeat, showing that color plays an important role in the model's ability to understand the scene.

Table 6: A comparison of tested S3E VLM instances in photorealistic blocksworld with different conflicts removed from the dataset. "rubber only" and "metal only"' limit to a single material, "color conflict" means no 2 shapes have the same color, and "color-size/color-shape conflict" means objects may have the same color if they don't share the same size/shape. Compared metrics are accuracy (3 thresholds $\theta$) and AP scores.

| | $\theta = 0.3$ | $\theta = 0.5$ | $\theta = 0.7$ | AP Score (micro) | AP Score (macro) |
|---|---|---|---|---|---|
| 0.5B (rubber only) | 0.58 | 0.72 | 0.76 | 0.38 | 0.76 |
| 0.5B (metal only) | 0.71 | 0.80 | 0.80 | 0.40 | 0.73 |
| 0.5B (color conflict) | 0.70 | 0.80 | 0.80 | 0.44 | 0.78 |
| 0.5B (color-size conflict) | 0.72 | 0.84 | 0.86 | 0.33 | 0.63 |
| 0.5B (color-shape conflict) | 0.72 | 0.86 | 0.88 | 0.28 | 0.56 |
| 0.5B + Instruct (rubber only) | 0.65 | 0.75 | 0.75 | 0.37 | 0.74 |
| 0.5B + Instruct (metal only) | 0.68 | 0.79 | 0.80 | 0.35 | 0.72 |
| 0.5B + Instruct (color conflict) | 0.72 | 0.80 | 0.79 | 0.41 | 0.77 |
| 0.5B + Instruct (color-size conflict) | 0.75 | 0.85 | 0.86 | 0.29 | 0.62 |
| 0.5B + Instruct (color-shape conflict) | 0.74 | 0.87 | 0.88 | 0.24 | 0.55 |
| 7B (rubber only) | 0.74 | 0.77 | 0.79 | 0.66 | 0.95 |
| 7B (metal only) | 0.78 | 0.81 | 0.82 | 0.68 | 0.93 |
| 7B (color conflict) | 0.79 | 0.80 | 0.82 | 0.67 | 0.94 |
| 7B (color-size conflict) | 0.77 | 0.80 | 0.83 | 0.59 | 0.85 |
| 7B (color-shape conflict) | 0.77 | 0.80 | 0.83 | 0.55 | 0.83 |
| 7B + Instruct (rubber only) | 0.77 | 0.81 | 0.83 | 0.71 | 0.93 |
| 7B + Instruct (metal only) | 0.83 | 0.84 | 0.84 | 0.73 | 0.90 |
| 7B + Instruct (color conflict) | 0.82 | 0.84 | 0.85 | 0.71 | 0.92 |
| 7B + Instruct (color-size conflict) | 0.81 | 0.84 | 0.86 | 0.62 | 0.83 |
| 7B + Instruct (color-shape conflict) | 0.80 | 0.84 | 0.87 | 0.57 | 0.81 |
| 72B (rubber only) | 0.87 | 0.88 | 0.89 | 0.83 | 0.96 |
| 72B (metal only) | 0.90 | 0.91 | 0.91 | 0.90 | 0.95 |
| 72B (color conflict) | 0.91 | 0.92 | 0.92 | 0.89 | 0.98 |
| 72B (color-size conflict) | 0.87 | 0.88 | 0.89 | 0.79 | 0.90 |
| 72B (color-shape conflict) | 0.85 | 0.87 | 0.88 | 0.74 | 0.87 |
| 72B + Instruct (rubber only) | 0.88 | 0.89 | 0.90 | 0.91 | 0.97 |
| 72B + Instruct (metal only) | 0.89 | 0.91 | 0.91 | 0.94 | 0.96 |
| 72B + Instruct (color conflict) | 0.91 | 0.92 | 0.93 | 0.94 | 0.98 |
| 72B + Instruct (color-size conflict) | 0.83 | 0.85 | 0.87 | 0.86 | 0.91 |
| 72B + Instruct (color-shape conflict) | 0.80 | 0.82 | 0.85 | 0.82 | 0.89 |

Using additional natural language instructions, we were able to mitigate this differentiation issue in the 72B model[2]. We used the following instructions:
*You will be asked questions about the state of blocks in a given image.*
*A block can be a cube, cylinder, or sphere.*
*A block is considered on the table if it is not on top of any other block.*
*Blocks come in one of two materials, rubber and metal. Rubber blocks have a matte finish while metal objects are glossy and reflective.*
We see an ~8% improvement for the 10 objects limit and ~1-3% improvement for all other sizes. When evaluated using the different object subsets, the instructed model is able to push performance even further, with 1-4% improvement, even though the performance was already relatively high without instruction. As in experiment 1, additional instruction only confuses the smaller models.

---

[2]due to time constraints, we were only able to run this model on about 7,000 data points. This should prove insignificant but will be amended in the final version of the paper.

## D  DATA COLLECTION

To collect the data points for our experiments (described in Section 6), we adhere to the following procedure:

1. Upon reset, the robot is set to the "home" position, and the groceries are randomly placed upright on one of the three tables.

2. An applicable action is chosen and executed by the robot.

3. If the action is completed successfully (target robot configuration achieved), the environment is rendered and the renderings are saved alongside the ground-truth task state.

4. If the action fails, a new action is sampled. The environment is reset after 5 consecutive failed attempts (step 1).

5. A new action is selected for execution (step 2). After 20 successful action executions, the environment is reset (step 1).

## E  PROMPTS

### E.1  PDDL PREDICATES TRANSLATION

To translate PDDL predicates to natural language questions, we use the following system prompt to instruct the LLM.

*The following is a PDDL domain*
*{DOMAIN}*
*Here are the names of all the objects in the current problem, sorted by their type:*
*{OBJECTS_BY_TYPE}*
*Given a grounded predicate with concrete variables, write a natural language yes-no query whose answer determines the truth value of the predicate.*
*Respond only with this natural language query and nothing else.*

The *DOMAIN* variable is the string description of the entire domain. In our case, this was the content of the PDDL domain. The *OBJECTS_BY_TYPE* variable is a comma-separated list of strings of the form:
*{OBJECT_TYPE_NAME} type: [{OBJECT1_NAME},{OBJECT2_NAME},...]*
where the *OBJECT_TYPE_NAME* and *OBJECTi_NAME* variables are the names as they appear in the PDDL domain file (for the types) and problem file (for the objects).

With this system prompt, the LLM is given a user prompt of the form:
*{PREDICATE}({VARIABLE1},{VARIABLE2},...)*
where the *PREDICATE* is a predicate from the PDDL domain file and *VARIABLEi* are objects from the PDDL problem file whose types match the predicate's variables. We do this for all ground predicates and create a mapping from predicates to their corresponding natural language query.

### E.2  VQA MODEL PROMPTS

The following is the system prompt used to calibrate the VQA model for state estimation:
*A curious human is asking an artificial intelligence assistant yes or no questions.*
*The assistant answers with one of three responses: YES or NO.*
*The assistant's response should not include any additional text.*

To estimate the value of a predicate given an array of images:

*{IMAGE_TOKEN}*
*{IMAGE_TOKEN}*
*...*
*{PREDICATE_NL_QUERY}*

*IMAGE_TOKEN* is a placeholder that is later replaced by the image representation of the VQA model, and *PREDICATE_NL_QUERY* is the input predicate's natural language form obtained from a LLM using the prompts described in Appendix E.1. The number of image tokens corresponds to the number of input images.

### E.3 ADDITIONAL INSTRUCTION PROMPTS

Additional instructions were appended to the end of the VQA model's system prompt. In experiment 1 described in Section 6, the instructions were as follows:

*The user will show you images of a simulated robot and ask questions about the state of the environment.*
*The milk carton is a clean white rectangular box with a triangular top.*
*When the robot is holding the milk carton it looks like there is a white rectangular object being pinched by the robot's gripper.*
*The red can of soda is a small red cylinder.*
*When the robot is holding the red can of soda it looks like there is a small red object that is enveloped by the robot's gripper.*
*The loaf of bread looks like a small brown box.*
*When the robot is gripping the loaf of bread it looks like there is a small brown object inside the robot gripper.*

## F  HARDWARE SPECIFICATIONS

We used three kinds of GPU models for our experiments. The Nvidia GeForce RTX 2080 Ti was our low performance GPU, with less than 11GB of memory. The Nvidia GeForce RTX 3090 was our mid-range performance GPU, with 24GB of memory. The Nvidia RTX A6000 was our high performance GPU, with 48GB of memory. The GPUs were operated using Intel Xeon Platinum 8180 CPUs. The machines were running Ubuntu 22.04.4 LTS with kernel version 5.15.0-119-generic.

We use the 70 Billion parameter LLaMA 3 model for predicate translation. This is a heavy model that requires 8 GeForce RTX 3090 (24GB). While this is a heavy requirement, the translation stage is executed in the preprocessing stage, and must only run once before running any number of times using the same translation. The 0.5B and 7B OV models can both run on a single GeForce RTX 2080 Ti (¡11GB) and GeForce RTX 3090 (24GB), respectively. The 72B OV was run with 4 RTX A6000 (48GB).

## G  PER PREDICATE AP SCORES

Fig. 9 shows the AP scores for each predicate in experiment 1 (simulated grocery sorting) individually. The results reveal which items are less recognized by the used VLM.

## H  ACRONYMS

**AP**  Average Precision. 8–10, 15–18, 20, 22

**CWA**  Closed World Assumption. 1

**LLaMA**  Large Language model Meta AI. 8, 10, 20
**LLaVA**  Large Language and Vision Assistant. 8, 10, 14
**LLM**  Large Language Model. 3–5, 8, 14, 19, 20

**OV**  OneVision. 8, 10, 20

**PDDL**  Planning Domain Definition Language. 3, 7, 8, 15, 19

**S3E**  Semantic Symbolic State Estimation. 1–10, 15–18, 22

**TMP** Task and Motion Planning. 2, 3

**VLM** Vision-Language Model. 1–3, 5–10, 14–18, 20, 22

**VQA** Visual Question Answering. 3–5, 7, 8, 10, 19, 20

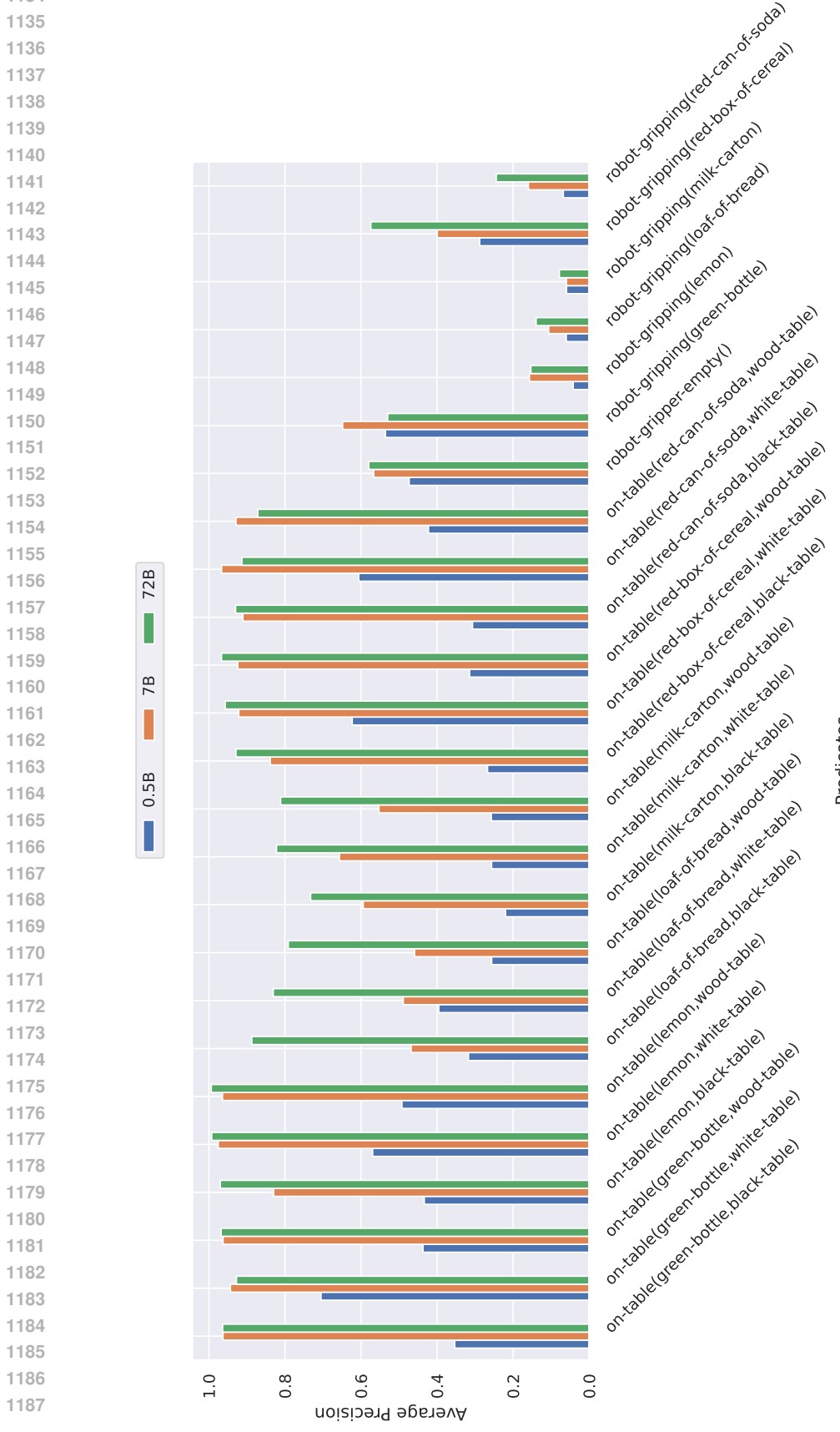

Figure 9: A comparison of tested S3E VLM sizes in experiment 1 (simulated) on AP score for each predicate separately.

