# OpenReview forum: "S3E: Semantic Symbolic State Estimation With Vision-Language Foundation Models"
_ICLR.cc/2025/Conference — Submitted to ICLR 2025_

### Official Review · Reviewer_TVcW · 2024-10-25

**Soundness:** 2
**Presentation:** 2
**Contribution:** 1
**Rating:** 3
**Confidence:** 4

**Summary:**

This paper proposed a state estimator that leverages Vision Language Models to translate sensor data into high-level task states. However, this paper provides limited technical contribution or insight, as it is a straightforward application of VLMs.

**Strengths:**

The paper is easy to understand.

The method is evaluated with both simulation and real-world images.

**Weaknesses:**

This paper is a straightforward application of VLMs. There is already a lot of work in robotics using VLM for state estimation, and some use symbolic representation to work with existing task and motion planning algorithms. [1,2,3]. These existing works often with much more complicated scenarios and systems than the ones demonstrated in this work. I don't think this provides much new insight or knowledge.

The paper does not provide sufficient discussion on related work.

The scenarios tested are very simple: only simple rigid objects on a tabletop setting with simple spatial relationships. "state estimator" should provide significantly richer information than just simple object class and their spatial relations.

The state estimation is not used or evaluated in realistic robotics tasks or systems. It only performs offline evaluation.

[1] Manipulate-Anything: Automating Real-World Robots using Vision-Language Models”
[2]AHA: A Vision-Language-Model for Detecting and Reasoning over Failures in Robotic Manipulation
[3]REFLECT: Summarizing Robot Experiences for Failure Explanation and Correction

**Questions:**

The paper is clear, I don't think answering any question would significantly change my evaluation. I think here is a few questions to think about.
1) how to extend the state estimation to more general scenarios, e.g., non-rigid object, where the object state is beyond the location and spatial relationship?
 2) What are some existing works in robotics using VLM as state estimation? Summary their approach and limitations and see how the proposed work can address them.

---

> ### Author Response · Authors · 2024-11-28
>
> We thank the reviewer for their feedback. We respectfully disagree with the characterization of our work as a straightforward application of VLMs and would like to clarify several points:
>
> Our contribution extends beyond simply applying VLMs to robotics. S3E presents the first general-purpose framework for translating arbitrary symbolic predicates into natural language queries that can be evaluated by VLMs. While the demonstration scenarios may appear simple, the method itself is not limited to these scenarios - it can work with any domain that can be represented using symbolic predicates. This includes complex predicates describing non-rigid object states, temporal relationships, or abstract properties, provided they can be expressed in the domain description language.
>
> We acknowledge that we should better situate our work within the context of existing VLM applications in robotics, including the papers mentioned by the reviewer (please see the revised version). However, our approach differs from these works in an important way: rather than using VLMs for specific robotic tasks or failure detection, we provide a general framework for bridging symbolic task planning with visual perception through systematic predicate translation and evaluation. This enables S3E to work with any planning system that uses symbolic state representations.
>
> Our offline evaluation focused on establishing the reliability of the state estimation component in isolation. While integration with complete robotic systems is important future work, understanding the capabilities and limitations of the core state estimation mechanism provides valuable insights for such integration.
>
> Regarding the specific concerns:
>
> **Question:** how to extend the state estimation to more general scenarios, e.g., non-rigid object, where the object state is beyond the location and spatial relationship?
> **Response:** The apparent simplicity of our test scenarios was chosen for clear demonstration and validation of the core method. Nothing in our approach inherently limits it to rigid objects or spatial relationships. Any predicate that can be expressed symbolically (e.g., "is_folded(towel)", "contains_liquid(cup)", "is_locked(door)") can be translated and evaluated using our framework, as long as it is visually discernible.
>
> **Question:** What are some existing works in robotics using VLM as state estimation? Summary their approach and limitations and see how the proposed work can address them.
> **Response:** We agree that expanding the related work discussion and demonstrating S3E in more complex scenarios would strengthen the paper. However, we believe our contribution provides a valuable foundation for developing more sophisticated symbolic state estimators that can leverage the continuing advances in vision-language models.
>
> We hope that our responses have shifted the reviewers perspective on our paper and have encouraged a reevaluation through of the revised submission.

---

### Official Review · Reviewer_ZVEE · 2024-10-29

**Soundness:** 2
**Presentation:** 2
**Contribution:** 2
**Rating:** 3
**Confidence:** 4

**Summary:**

The paper addresses the problem of estimating the truth values of symbolic state predicates, which is essential for PDDL-style planning. While prior works has performed state estimation in a task-specific setting, the proposed approaches uses foundation models as universal estimator that can be conditioned on the task through prompts. First, natural language question is created for every for every possible grounded predicates in the given domain. The translated question along with the image is passed to a VQA model (a VLM) to get the truth value of the predicate. The evaluation is done on a table top manipulation environment which is fully observable and uncluttered, which shows that the proposed method can ground the symbolic predicates across different domain/tasks.

**Strengths:**

- The paper attempts to address an important problem for semantic state estimation.
- The consideration of both the predicates and their uncertainty is a promising direction.
- The evaluation is at a task level showing utility of the approach for realistic applications.

**Weaknesses:**

- The proposed approach of determining the validity of predicates with VQA addresses only a part of the problem of estimating the symbolic state. The proposed approach is restricted to simple predicates in small clutter free settings. Scalability to scenarios with complex object arrangements and to higher-order predicates is limited.
- The problem of eliciting uncertainty from LLMs has received attention in the past. The authors are requested to highlight the technical gap they are filling in this regard. Please refer to works such as https://arxiv.org/pdf/2307.01928 and https://arxiv.org/pdf/2306.13063.
- The problem of inferring semantic properties of the robot's environment has also been addressed in works that extract a scene graph from raw sensor data. Please see Maggio et al. for an example (https://arxiv.org/abs/2404.13696). I request authors to situate their work along side such efforts.

**Questions:**

1. Scalability of exhaustive enumeration. The translation stage of each predicate query involves posing predicate truth value assessment as answer to a textual question. It appears from the text that this conversion is done for each predicate independently and exhaustively. However, there may be correlation in the predicates. If the robot is far away from the table then it is far away from all blocks on the table. Does the approach take advantage of such logical connections?

2. Agent-dependent predicates. Robot interaction tasks require predicates such as in_hand, robot_close_to etc. predicates. How is the agent morphology and other aspects communicated to the VLM for accurate estimation of these predicates.

3. Higher order predicates. Often in task planning there is a need for higher order predicates. For example, consider a stack of blocks. One would need to reason about the top of the stack as different from a block at the base. See Pasula et al. (https://people.csail.mit.edu/lpk/papers/zpk.pdf) for an example of such predicate (topstack()). Such predicates requires higher order reasoning over the atomic properties that may hold in the scene. Does the proposed method address computation of such predicates.

4. Baseline VLM performance for direct plan synthesis. The authors have taken the symbolic planning view where knowledge of state predicates is fundamental. Alternatively, what if one asks the VLM to directly synthesise robot actions for a given goal without explicitly querying for the predicates. It would be insightful to analyse this approach as a baseline.

**Details Of Ethics Concerns:**

None.

---

> ### Author Response · Authors · 2024-11-28
>
> Thank you for your feedback and for recognizing the importance of our work in semantic state estimation. We address your main concerns below:
>
> **Concern:** The proposed approach of determining the validity of predicates with VQA addresses only a part of the problem of estimating the symbolic state. The proposed approach is restricted to simple predicates in small clutter free settings. Scalability to scenarios with complex object arrangements and to higher-order predicates is limited.
> **Response:** We agree that our current evaluation focuses on relatively simple predicates in clutter-free environments. Though one might argue that photorealistic blocksworld with 10 objects can become cluttered at times, using a clean and clear environment was a deliberate choice for the initial demonstration of S3E's capabilities. Our results show that, there are several issues to overcome before we can tackle more complex environments. Specifically, the presented uncertainties and limitations that come bundled with using VLMs directly as state estimators must be addressed. Some of these issues will improve as general VLM quality improves, and we have suggested the rest as future work.
>
> **Concern:** The problem of eliciting uncertainty from LLMs has received attention in the past. The authors are requested to highlight the technical gap they are filling in this regard. Please refer to works such as [Robots That Ask For Help](https://arxiv.org/pdf/2307.01928) and [Can LLMs Express Their Uncertainties](https://arxiv.org/pdf/2306.13063).
> **Response:** We appreciate you pointing out these relevant works on uncertainty elicitation in LLMs. While these works advance the field, they primarily focus on different problem settings. They aim to calibrate the confidence scores produced by LLMs to better reflect their actual accuracy while other approaches focus on detecting and mitigating overconfidence in LLM responses, particularly in conversational settings. The technical gap we address is leveraging these advancements for symbolic state estimation within a task planning context. Our work bridges the domains of symbolic planning and VLM-based uncertainty quantification, exploring how to utilize uncertainty information to improve state estimation and ultimately task success. We will clarify this distinction in the revised paper and include a more comprehensive discussion of related work on LLM uncertainty.
>
> **Concern:** The problem of inferring semantic properties of the robot's environment has also been addressed in works that extract a scene graph from raw sensor data. Please see [Maggio et al.](https://arxiv.org/abs/2404.13696) for an example. I request authors to situate their work along side such efforts.
> **Response:** We agree that scene graph extraction techniques offer a valuable approach for understanding the robot's environment. However, our work differs in its focus and objective. Scene graphs aim to represent the relationships between objects in a scene comprehensively. They are often used for tasks like image captioning or visual question answering. S3E, on the other hand, focuses specifically on estimating the truth values of symbolic predicates relevant to a given task. We use VLMs as a tool to efficiently assess these predicates without needing to construct a full scene graph, which can be computationally expensive. In the revised paper, we clarify this distinction and discuss the complementary nature of S3E and scene graph extraction methods. We will also suggest ways of integrating these approaches to further improve the robustness and accuracy of symbolic state estimation.

---

> > ### Author Response · Authors · 2024-11-28
> >
> > Regarding the specific questions:
> >
> > **Question:** Scalability of exhaustive enumeration. The translation stage of each predicate query involves posing predicate truth value assessment as answer to a textual question. It appears from the text that this conversion is done for each predicate independently and exhaustively. However, there may be correlation in the predicates. If the robot is far away from the table then it is far away from all blocks on the table. Does the approach take advantage of such logical connections?
> > **Response:** The reviewer raises an excellent point about predicate correlations. Currently, our method evaluates each predicate independently, which does not take advantage of logical relationships between predicates (e.g only one object can be held by robot). This represents an opportunity to improve our method by leveraging the language model to identify and utilize such correlations. We have addressed this limitation in the revised version of the paper and suggest it as future work.
> >
> > **Question:** Agent-dependent predicates. Robot interaction tasks require predicates such as in_hand, robot_close_to etc. predicates. How is the agent morphology and other aspects communicated to the VLM for accurate estimation of these predicates.
> > **Resopnse** For agent-dependent predicates like "in_hand" or "robot_close_to", our approach relies on the VLM's inherent reasoning capabilities rather than explicit communication of robot morphology. While this worked sufficiently in our experiments, we acknowledge that more complex robot configurations might require explicit communication of agent properties to the VLM. While we did not test such settings, we believe the issue can be mitigated using the same approach of natural language instruction. We leave it as future work to to incorporate robot morphology in a more sophisticated manner.
> >
> > **Question:** Higher order predicates. Often in task planning there is a need for higher order predicates. For example, consider a stack of blocks. One would need to reason about the top of the stack as different from a block at the base. See [Pasula et al.](https://people.csail.mit.edu/lpk/papers/zpk.pdf) for an example of such predicate (topstack()). Such predicates requires higher order reasoning over the atomic properties that may hold in the scene. Does the proposed method address computation of such predicates.
> > **Response:** Our current implementation focuses on predicates explicitly defined in the domain description. We acknowledge that higher-order predicates requiring reasoning over multiple atomic properties are not directly addressed in our current approach. While VLMs might be capable of such reasoning, extending S3E to handle higher-order predicates would require additional research on how to effectively translate and query such properties.
> >
> > **Question:** Baseline VLM performance for direct plan synthesis. The authors have taken the symbolic planning view where knowledge of state predicates is fundamental. Alternatively, what if one asks the VLM to directly synthesise robot actions for a given goal without explicitly querying for the predicates. It would be insightful to analyse this approach as a baseline.
> > **Response:** While using VLMs directly for plan synthesis is an interesting direction that has received significant attention in recent work, our paper specifically focuses on improving the state estimation module of AI agents. We intentionally kept the planning aspect separate to ensure our contribution could benefit any planning approach, whether classical or learning-based, without introducing bias from the chosen planner or learning algorithm. This modularity allows S3E to be integrated with various planning methods while maintaining a clear focus on the state estimation challenge.
> >
> > We hope our responses address all the reviewers concerns and questions.

---

### Official Review · Reviewer_nmf7 · 2024-11-04

**Soundness:** 2
**Presentation:** 3
**Contribution:** 2
**Rating:** 6
**Confidence:** 4

**Summary:**

This paper introduces S3E, a novel approach that uses vision-language foundation models for automated state estimation in robotic task planning. While traditional approaches require hand-crafted state estimation functions for each task, S3E offers a zero-shot solution that translates symbolic predicates into natural language questions, uses vision-language models to answer these questions from visual input, and converts the answers back into symbolic states. The system's effectiveness is validated through experiments in grocery sorting tasks and photorealistic blocksworld domains, particularly when employing uncertainty mitigation strategies through natural language instruction and environment design.

**Strengths:**

The key strength of this paper lies in its practical approach to solving a significant challenge in robotics - automated state estimation. The authors present a solution that eliminates the need for hand-crafted state estimators, traditionally a major bottleneck in deploying robotic systems across different tasks. The authors propose an approach that bridges symbolic task planning with modern vision-language models through natural language as an intermediate representation. They provide practical mitigation strategies through natural language instruction and environment design, demonstrating strong empirical results with over 90% precision.

**Weaknesses:**

There are some major problems that concerns me:
1. While S3E is presented as a zero-shot solution, it still requires significant assumptions about the environment, including full observability, visually distinguishable objects, and unambiguous domain descriptions - constraints that may not hold in many real-world applications.

2. The experiments, while showing promising results, are limited to relatively controlled environments and simple manipulation tasks. The evaluation could be strengthened by exploring more challenging scenarios such as partial observability conditions, dynamic environments, or multi-agent interactions. This would help better understand the method's full potential.

3. The paper's handling of uncertainty, while acknowledged, could be more rigorous - the proposed mitigation strategies of environment design and natural language instruction essentially sidestep rather than solve the underlying uncertainty challenges. Additionally, the method's heavy dependence on the quality of the vision-language model's training data suggests potential brittleness when encountering out-of-distribution scenarios, as evidenced by the poor performance with certain objects in the simulated environment.

**Questions:**

1. Have you explored any cases where the domain descriptions contain partial or ambiguous information? What were the results?
2. Were there any failure cases in the controlled environments that could provide insights into the system's limitations?

---

> ### Author Response · Authors · 2024-11-28
>
> We appreciate the reviewer's thorough analysis and constructive feedback. We address each weakness and the questions below:
>
> **Concern:** While S3E is presented as a zero-shot solution, it still requires significant assumptions about the environment, including full observability, visually distinguishable objects, and unambiguous domain descriptions - constraints that may not hold in many real-world applications.
> **Resonse:** We agree that the assumptions of full observability, visual distinguishability, and unambiguous domain descriptions represent significant constraints. These challenges are fundamental to state estimation in general, not just our approach. In our paper, we explicitly analyze how these assumptions affect S3E's performance (as shown in Tables 5-6 and discussed in Section 6.2), demonstrating both the capabilities and current limitations of our method. This transparency about performance limitations under different conditions helps establish a baseline for future work in this area.
>
> **Concern:** he experiments, while showing promising results, are limited to relatively controlled environments and simple manipulation tasks. The evaluation could be strengthened by exploring more challenging scenarios such as partial observability conditions, dynamic environments, or multi-agent interactions. This would help better understand the method's full potential.
> **Response:** We acknowledge that exploring more challenging scenarios would provide valuable insights. Our work represents a first step in demonstrating the viability of using VLMs for state estimation. Note, however, that while results show promise, these controlled environments have already shed light on the limitations of VLMs as state estimators, e.g., out-of-distribution observations, setting the foundation for important future work.
>
> **Concern:** The paper's handling of uncertainty, while acknowledged, could be more rigorous - the proposed mitigation strategies of environment design and natural language instruction essentially sidestep rather than solve the underlying uncertainty challenges. Additionally, the method's heavy dependence on the quality of the vision-language model's training data suggests potential brittleness when encountering out-of-distribution scenarios, as evidenced by the poor performance with certain objects in the simulated environment.
> **Response:** The reviewer makes a valid point about our handling of uncertainty. Our goal was to first identify and analyze these uncertainties rather than solve them completely. While our mitigation strategies may appear to sidestep the challenges, they provide practical solutions for immediate deployment through an iterative process between the user and the VLM while highlighting areas needing further research. Regarding the dependence on VLM training data quality, we view this as both a limitation and an opportunity - as VLM technology improves, our method automatically benefits from these advances without requiring modifications to the core approach.
>
> Regarding the specific questions:
>
> **Question:** Have you explored any cases where the domain descriptions contain partial or ambiguous information? What were the results?
> **Response:** Yes, we did test the case of ambiguous information in the domain description. Specifically, the ambiguity presented in Figure 4 arises from ambiguity in the meaning of "is gripping". This is addressed using the "Pose" modification. Furthermore, the photorealistic blocksworld domain presented in Appendix C uses a variation of the "on" predicate, leaving room for interpretation of what "block A is on block B" means.
>
> **Question:** Were there any failure cases in the controlled environments that could provide insights into the system's limitations?
> **Response:** Yes, we explicitly documented failure cases in our paper. For example, in Section 6.2 and Figure 5, we discuss how certain objects (like the bread) were consistently misrecognized even with natural language instruction, highlighting limitations in the VLM's ability to handle visually unrealistic objects. Additionally, Figure 4 shows an example where object gripping state was ambiguous, demonstrating how uncertainties can arise even in controlled environments.
>
> We hope that our responses satisfy the reviewers concerns and curiosities about our approach.

---

### Official Review · Reviewer_hVcq · 2024-11-05

**Soundness:** 2
**Presentation:** 2
**Contribution:** 2
**Rating:** 3
**Confidence:** 4

**Summary:**

This paper introduces the use of pre-trained VLM to classify symbolic states. Specifically, a symbolic predicate is first converted to language instruction and is then fed to a VLM together with the image; the token probabilities of the "yes" and "no" answers are used to determine the probability of the predicate state being true. The paper proposes to deal with two types of uncertainties that arise from partial observation and ambiguities in the definitions of the symbolic predicate. To combat this first type of uncertainty, the authors propose manipulating and moving the occluded objects to be in direct sight. For the second type of uncertainty, the authors propose to provide clearifying instructions.

**Strengths:**

Traditional methods for classifying symbolic states usually need to be manually designed by domain experts or trained with a large amount of data. This paper investigates using pre-trained VLMs to classify states in zero-shot. This approach can potentially improve the generalizability of task-planning methods.

**Weaknesses:**

1. It's unclear how the system automatically determines that there is uncertainty in the prediction, which then needs to be addressed by asking for clarifying instructions or better camera views.
2. The predicates this paper tests are limited to determining whether an object is on a specific table region or whether the object is in hand or in the air. These predicates probably can be determined using very simple heuristics based on gripper state or detected object bounding boxes. It's hard to assess whether the method can reliably classify more diverse predicates for real-world robotic tasks. I suggest the authors to look at other photo-realistic simulation environments that provide annotated object states.
3. Even though the authors motivate the paper for task planning, there is no evaluation of the method integrated with a task planning framework. It's unclear how the improvement in classification performance translates to the improvement in overall task success.
4. In line 176, the authors mention that the assumption is unrealistic. However, I think this assumption is valid if the action sequences are correct (e.g., if the robots are being teleoperated by a human user)
5. In general, the writing of the paper can also be improved. I found the discussion on uncertainty and experiment section a little confusing.

**Questions:**

I hope the authors can address my concerns above. I don't have additional questions.

---

> ### Author Response · Authors · 2024-11-28
>
> We thank the reviewer for their thoughtful feedback. Below we address each concern:
>
> **Concern:** It's unclear how the system automatically determines that there is uncertainty in the prediction, which then needs to be addressed by asking for clarifying instructions or better camera views.
> **Response:** We appreciate this observation about uncertainty detection. To clarify, our paper's scope was to identify and analyze two types of uncertainties that emerge when using VLMs for state estimation, rather than to propose an automated system for detecting and resolving these uncertainties. We quantified these uncertainties through our experimental results and demonstrated their impact on state estimation performance. The development of automated uncertainty detection and resolution methods represents an important direction for future work. Currently, the calibration of the system is an iterative process between the user and the VLM. This is to be expected anyway because specific usage often leads to out-of-distribution examples. Should we replace S3E with a human, we would still need to explain these examples and make sure they understand before we develop trust.
>
> **Concern:** The predicates this paper tests are limited to determining whether an object is on a specific table region or whether the object is in hand or in the air. These predicates probably can be determined using very simple heuristics based on gripper state or detected object bounding boxes. It's hard to assess whether the method can reliably classify more diverse predicates for real-world robotic tasks. I suggest the authors to look at other photo-realistic simulation environments that provide annotated object states.
> **Response:** While the real-robot demonstration used relatively simple predicates for clarity, our evaluation included more complex scenarios through our simulated environments. In particular, our evaluation of the photorealistic blocksworld domain (Appendix C) tested the system with diverse object attributes including size, color, material, and shape, along with spatial relationships between objects. The success of S3E in handling these more complex predicates suggests that the method can scale to more sophisticated real-world scenarios, limited primarily by the underlying VLM's generalization capabilities.
>
> **Concern:** Even though the authors motivate the paper for task planning, there is no evaluation of the method integrated with a task planning framework. It's unclear how the improvement in classification performance translates to the improvement in overall task success.
> **Response:** We acknowledge that evaluating S3E's impact on overall task success would provide valuable insights. However, such evaluations insert bias through the choice of planner, the strategy for sampling states, and the stochasticity of irreversible action failure. Our work focused specifically on improving the state estimation module, which is one critical component of a complete robotic system. While we demonstrate good state estimation accuracy, evaluating how this accuracy translates to overall task success when integrated with planning and control modules represents important future work.
>
> **Concern:** In line 176, the authors mention that the assumption is unrealistic. However, I think this assumption is valid if the action sequences are correct (e.g., if the robots are being teleoperated by a human user)
> **Response:** We respectfully disagree about the assumption in line 176. While action sequences may be reliable under human teleoperation, our work targets autonomous operation in unstructured real-world environments where action outcomes can be uncertain due to various factors (e.g., environmental dynamics, sensor noise, control errors). In such scenarios, reliable state estimation and replanning capabilities are crucial for robust task execution.
>
>
> **Concern:** In general, the writing of the paper can also be improved. I found the discussion on uncertainty and experiment section a little confusing.
> **Response:** We appreciate the feedback about the clarity of our uncertainty discussion and experimental section. In the revised version, we improved the organization and presentation of these sections to make our methodology and results more accessible. Specifically, we restructured Sections 5 to more clearly delineate between the two types of uncertainties and their implications.
>
> We believe these clarifications address the reviewer's main concerns while acknowledging limitations and future work directions.

---

### Author Response · Authors · 2024-11-28
**Summary of Revisions**

We are grateful to all reviewers for the insightful reviews. We address the specific issues raised by each reviewer individual comments. Here, we  summarize the main changes and additions made to the paper below:

- We address related additional related work suggested by the reviewers. This is reflected in the "Introduction" and the "Related Work" sections. The newly introduced work includes
	- eliciting uncertainties from language modles
	- scene graphs
	- additional work that use VLMs for state estimation
- We distinguish and elaborate on the different kinds of uncertainties in section 5 (Uncertainty in Semantic State Estimation)
- We reorganized the results (section 6.2) to make our results more accessible.
- We made several clarifications in the Conclusion (section 7) to further emphasize the limitations of S3E and our suggested avenues for future work.

We believe that this revision addresses the reviewers' main concerns.

---

### Meta-Review · Area_Chair_xaH6 · 2024-12-16

**Metareview:**

The paper proposes Semantic Symbolic State Estimation (S3E), which uses pre-trained VLMs to determine the truth value of symbolic state predicates from visual observations. S3E creates a natural language question for each grounded predicate. The resulting question together with the image are passed to a VLM to determine the probability of the predicate being true. The paper analyzes different types of uncertainties that can arise, e.g., due to partial observability. The paper evaluates S3E usings a table-top manipulation domain.

The paper was reviewed by four referees who largely agree on its primary strengths and weaknesses. Several reviewers appreciate the benefits of using a a pre-trained VLM for state estimation, particularly when compared to traditional methods that require the involvement of domain experts or the need for extensive training data. In this regard, a key strength of the paper is in providing a potentially practical approach to state estimation, which demonstrates strong empirical results on the domains that are considered. However, as several reviewers pointed out, the evaluation considers a limited set of predicates with simplistic scenarios that involve little-to-no clutter and full observability. The reviewers question whether a simple heuristic would prove effective in these settings and raise doubts about whether the method could be extended to more challenging and practical settings. Meanwhile, at least two reviewers find that a more complete discussion of related work on semantic state estimation and uncertainty quantification is needed to clarify the paper's contributions. The AC appreciates the effort that the authors put into responding to the initial reviews, which included an honest acknowledgement of some of the paper's limitations.

**Additional Comments On Reviewer Discussion:**

No discussion was necessary.

---

### Decision · Program_Chairs · 2025-01-22

Reject